# Rural–Urban Disparities in Patient Care Experiences among Prostate Cancer Survivors: A SEER-CAHPS Study

**DOI:** 10.3390/cancers15071939

**Published:** 2023-03-23

**Authors:** Ambrish A. Pandit, Nilesh N. Patil, Mostafa Mostafa, Mohamed Kamel, Michael T. Halpern, Chenghui Li

**Affiliations:** 1Division of Pharmaceutical Evaluation and Policy, University of Arkansas for Medical Sciences (UAMS), Little Rock, AR 72205, USA; 2Department of Surgery, College of Medicine, University of Cincinnati, Cincinnati, OH 45221, USA; 3Asiut University Hospitals and School of Medicine, Asiut 2074020, Egypt; 4Department of Urology, Ain Shams University, Cairo 11566, Egypt; 5Healthcare Delivery Research Program, Division of Cancer Control and Population Sciences, National Cancer Institute, Bethesda, MD 20892, USA

**Keywords:** SEER-CAHPS, prostate cancer, rural–urban disparities, patient care experiences, definitive treatment

## Abstract

**Simple Summary:**

Rural residence has been associated with poor access to healthcare and accordingly lower odds of receiving definitive treatment. Patient care experiences are important indicators of the quality of care delivered and are highly valued by patients. Although rural–urban disparities in prostate cancer care are evident, it is unknown how these disparities are associated with PCEs among PCa survivors. In this retrospective cohort study, which included 3379 older PCa survivors at intermediate-to-high risk of disease progression, we evaluated the rural–urban differences in patient care experiences according to receipt of definitive treatment. We found rural PCa survivors were less likely to receive treatment. Treated rural PCa survivors reported better access to care, while untreated rural PCa survivors reported poorer care access and experiences compared to their large metro counterparts. We also explored rural–urban disparities in receipt of definitive treatment across different geographic regions. This study highlights the importance of conducting subgroup analysis to uncover any important heterogeneous care experiences among cancer survivors.

**Abstract:**

Background: We sought to evaluate rural–urban disparities in patient care experiences (PCEs) among localized prostate cancer (PCa) survivors at intermediate-to-high risk of disease progression. Methods: Using 2007–2015 Surveillance Epidemiology and End Results (SEER) data linked to Medicare Consumer Assessment of Healthcare Providers and Systems (CAHPS) surveys, we analyzed survivors’ first survey ≥6 months post-diagnosis. Covariate adjusted linear regressions were used to estimate associations of treatment status (definitive treatment vs. none) and residence (large metro vs. metro vs. rural) with PCE composite and rating measures. Results: Among 3779 PCa survivors, 1798 (53.2%) and 370 (10.9%) resided in large metro and rural areas, respectively; more rural (vs. large metro) residents were untreated (21.9% vs. 16.7%; *p* = 0.017). Untreated (vs. treated) PCa survivors reported lower scores for doctor communication (ß = −2.0; *p* = 0.022), specialist rating (ß = −2.5; *p* = 0.008), and overall care rating (ß = −2.4; *p* = 0.006). While treated rural survivors gave higher (ß = 3.6; *p* = 0.022) scores for obtaining needed care, untreated rural survivors gave lower scores for obtaining needed care (ß = −7.0; *p* = 0.017) and a lower health plan rating (ß = −7.9; *p* = 0.003) compared to their respective counterparts in large metro areas. Conclusions: Rural PCa survivors are less likely to receive treatment. Rural–urban differences in PCEs varied by treatment status.

## 1. Introduction

Prostate cancer (PCa) is the most common cancer affecting men in the United States [1]. Globally, 1,276,000 new PCa cases were diagnosed, and 359,000 men suffered from PCa in 2018 [2]. A positive family history of PCa, in addition to age and African American ancestry, is the most significant risk factor for developing PCa [3]. The progression of PCa is often slow with a five-year survival rate of >95% [4]. About three-fourths of the individuals with PCa are diagnosed at the localized stage [4], and >60% of them can carry an intermediate or high risk of disease progression [5]. Surgery and radiation therapy are recommended for treatment of patients with intermediate or high risk PCa [6]. Several disparities have been noted in the quality of care delivered to PCa survivors. Specifically, rural residence has been associated with poor access to healthcare and accordingly lower odds of receiving definitive treatment [7,8,9]. Likewise, the National Academy of Medicine has identified rural location as a potential risk factor for health care disparity [10]. Rural PCa survivors face unique challenges in accessing care such as a lack of subspecialized urologists in rural areas [11], lower access to advanced imaging modalities (including a pre-biopsy MRI) [12], hospital closures due to consolidation [13], and a lower inclination of newly trained physicians to practice in rural areas [14]. Access to healthcare promptly and the quality of care are likely to affect cancer care outcomes such as mortality, although the evidence regarding the association of rural/urban residence with mortality is mixed [15,16,17] depending on the study setting and adjustment of other demographic and clinical characteristics.

Patient care experiences (PCEs) such as obtaining needed care, obtaining care quickly, good patient–provider communication, and ease of access to health-related information are important indicators of the quality of care delivered and are highly valued by patients [18]. Thus, PCEs provide insight into the quality of healthcare plans and the overall quality of healthcare delivered [19]. Although rural–urban disparities in prostate cancer care are evident [7,8,12], it is unknown how these disparities are associated with PCEs among PCa survivors. In this exploratory study, we addressed this gap in knowledge by evaluating the Surveillance Epidemiology and End Results (SEER) data linked to the Consumer Assessment of Healthcare Providers and Systems (CAHPS) database using the question: Are there rural–urban differences in PCEs of men with prostate cancer?

## 2. Materials and Methods

### 2.1. Study Dataset

This study used the 2007-2015 SEER-CAHPS [17]. SEER is a cancer registry that provides information on patient demographics as well as cancer-related clinical information such as the tumor primary site, tumor stage and morphology, first course of treatment, and follow-up for vital status [20]. CAHPS surveys Medicare enrollees about their demographics and healthcare experiences [21]. We further linked the United States Federal Information Processing Standards (FIPS) codes from CAHPS to the United States Department of Agriculture Economic Research Service Rural–Urban Continuum Codes (RUCCs) [22] to obtain the RUCCs at the time of the survey.

### 2.2. Study Population

The study population included PCa survivors with localized prostate cancer without lymph node involvement and an intermediate or high risk of disease. Risk of disease progression was defined according to the National Comprehensive Cancer Network guidelines [23]. We chose this population because definitive treatment with either surgery or radiation was recommended [6]. Among them, we included PCa survivors who completed CAHPS surveys ≥ 6 months after PCa diagnosis and whose rural–urban residence was the same at diagnosis and survey completion. Their first survey after PCa diagnosis (completed ≥ 6 months after diagnosis) was analyzed. We excluded individuals (a) with a missing month/year of PCa diagnosis, (b) a missing survey date, (c) who were diagnosed at autopsy or through death certificate, (d) without a valid score for any of the PCE measures, and (e) who were missing definitive treatment status. Definitive treatment was defined as the receipt of definitive surgery and/or radiation [24]. Definitive surgery was defined as radical prostatectomy, prostatectomy with resection in continuity with other organs, pelvic exenteration, or radiation treatment that consisted of either external beam radiation therapy or brachytherapy. Those who received neither surgery nor radiation were categorized as not having received definitive treatment. After applying all exclusions, the study sample was classified as (1) those who received definitive treatment (referred to as ‘treated’ hereafter), and (2) those who did not receive definitive treatment (referred to as ‘untreated’ hereafter).

Among the treated PCa survivors, we further excluded (a) those who did not have any CAHPS survey post-definitive treatment and (b) those who received definitive treatment more than 12 months after PCa diagnosis.

### 2.3. Outcome Variables

We used the five composite measures of PCE (“getting needed care”, “getting care quickly”, “physician communication”, “getting needed prescription drugs”, and “customer service”) and four single-item rating measures (“overall care”, “health plan”, “primary physician”, and “specialist physician”) from CAHPS as outcome variables. The composite scores ranged from 0 to 100, while the ratings for rating measures ranged from 0 to 10. We used a linear mean scoring method to convert ratings into an interval-level response ranging from 0 to 100 [25]. Consistent with previous research, differences in PCE scores that were less than 3 points were considered “small”, ≥3 but <5 points were considered “medium”, while ≥5 points were considered “large” differences [26].

### 2.4. Exposure Variables

Rural–urban residence was our primary exposure variable. RUCCs derived from county FIPS codes of survivors’ residences were used to classify rural–urban status into three categories: “Large Metro” (RUCC of 0 or 1), “Metro” (RUCC of 2 or 3), and “Rural” (RUCC from 4 to 9). Treatment status was our secondary exposure variable and was dichotomized into receipt or non-receipt of definitive treatment. We determined the associations between treatment status and PCEs.

### 2.5. Covariates

We adjusted all models using demographic and clinical covariates. SEER-CAHPS recommends adjusting for associations using case-mix variables when comparing PCEs across groups [25]. Among the recommended case-mix variables, we adjusted for age when participants responded to the survey, race (non-Hispanic white, non-Hispanic black, Hispanic, Asian, and other), education (some college or above, high school or less, or missing), general health status (excellent, very good, good fair, poor, or missing), mental health status (excellent, very good, good fair, poor, or missing), proxy answering questions for respondent (yes, no, or missing), dual eligibility (yes, no, or missing), and low-income subsidy (yes, no, or missing) because these variables were present in all the years of the study dataset. Additionally, we adjusted for plan type (Fee-For-Service and Medicare Advantage) [27], marital status (married, not married, or missing) [28], geographic region of residence at the time of the CAHPS survey (Northeast, Midwest, South, or West) [29], Census Tract Poverty Indicator (0% to <5% poverty, 5% to <10% poverty, 10% to <20% poverty, 20% to 100% poverty, or unknown) [28], survey year (2008, 2009, 2010, 2011, 2012, 2013, 2014, or 2015) [27], smoking status (non-smoker, smoker, or missing) [29], tumor grade (well-differentiated, moderately differentiated, poorly differentiated, undifferentiated, or unknown), risk (intermediate or high), number of prior cancers other than prostate cancer (0, 1, 2, or 3), comorbidity count (0, 1, 2, or 3–4) [29], and the time between PCa diagnosis and CAHPS survey (<2 years, 2 to 5 years, or >5 years) [27]. The CAHPS asks responders about 4 types of comorbidities: heart conditions, stroke, chronic obstructive pulmonary disease, and diabetes; thus, the maximum comorbidity count was 4.

### 2.6. Statistical Analysis

We compared the demographics and clinical characteristics of PCa survivors according to rural–urban residence categories using chi-squared tests and Fisher tests as appropriate for categorical variables and ANOVA tests for continuous variables. We determined the proportion of survivors treated as well as time to receive definitive treatment according to rural–urban residence.

We first performed multivariate linear regression modeling to evaluate the association of rural–urban residence with each PCE measure while adjusting for covariates and treatment status. Since the associations between rural–urban residence and PCE measures could differ according to receipt of definitive treatment, we additionally conducted separate analyses stratified by treatment status. For the analysis of the treated group, the regression model additionally adjusted for definitive surgery (yes, no, or missing) and radiation (yes, no, or missing); the time between PCa diagnosis and CAHPS survey was further broken down to the time between PCa diagnosis and definitive treatment (≤1 month, >1 to 2 months, >2 to 3 months, or >3 months) and time between definitive treatment and CAHPS survey (<1 year, 1 to <2 years, 2 to <3 years, or >3 years).

Rural areas in different geographic areas may face different issues in accessing healthcare [30,31,32]. As an exploratory analysis, we also performed unadjusted analyses to compare rural–urban disparities in the receipt of definitive treatment across different geographic regions. Among treated PCa survivors, we additionally performed unadjusted analyses to determine time-to-treatment disparities according to rural–urban residence in different geographic regions (Northeast, Midwest, South, or West). Only unadjusted analyses were conducted due to the small sample sizes.

We used SAS v.9.4 to perform the statistical analysis. This research was determined to be non-human subject research by the University of Arkansas for Medical Sciences Institutional Review Board (IRB # 260675).

## 3. Results

### 3.1. Study Cohort and Demographics

After applying inclusion and exclusion criteria, the study included 3379 PCa survivors (Figure 1). Among them, 1798 (53.2%), 1211 (35.8%), and 370 (10.9%) resided in large metro, metro, and rural areas, respectively.

Table 1 compares demographic and clinical factors according to rural–urban residence. Compared to metro and large metro PCa survivors, rural PCa survivors were less likely to have some college education or above (rural vs. metro vs. large metro: 46.0% vs. 56.7% vs. 59.5%; *p* < 0.001) and reside in the West region (rural vs. metro vs. large metro: 39.5% vs. 55.0% vs. 46.2%; *p* < 0.001), but more likely to reside in neighborhoods where 20–100% of residents were in poverty (rural vs. metro vs. large metro: 28.7% vs. 17.8% vs. 15.0%; *p* < 0.001). Interestingly, although a lower proportion of rural compared to large metro PCa survivors received definitive treatment (78.1% vs. 83.3%; *p* = 0.017), among those who received definitive treatment, a higher proportion of treated rural compared to large metro PCa survivors received definitive treatment within one month of diagnosis (42.2% vs. 31.3%; *p* < 0.001). Although the proportion of PCa survivors receiving definitive treatment did not differ significantly between metro and large metro residents, significantly more treated PCa survivors from metro areas vs. large metro (38.3% vs. 31.3%; *p* < 0.001) received it within one month of PCa diagnosis.

### 3.2. Regression Analyses

In the overall multivariate linear regression analysis in which we adjusted for rural–urban status and treatment status in addition to other covariates, we found that rural–urban status was not significantly associated with any of the PCEs, but receipt of definitive treatment was significantly associated with several PCEs after adjusting for covariates (including rural–urban residence). Table 2 provides the adjusted least-square mean estimates of the PCE scores (and standard errors (SEs)) and adjusted mean differences in the PCEs (and 95% confidence intervals (CIs)) according to treatment status from these regressions. The adjusted mean differences are beta coefficients from the linear regressions representing the mean differences in ratings or composite scores comparing untreated vs. treated patients. Compared to the treated survivors, the untreated PCa survivors reported lower scores for physician communication (ß = −2.0, 95% CI −3.8 to −0.2; *p* = 0.022), specialist rating (ß = −2.5, 95% CI −4.4 to −0.7; *p* = 0.008), and overall care rating (ß = −2.4, 95% CI −4.1 to −0.7; *p* = 0.006). The adjusted least-square mean estimates and differences according to rural–urban status are reported in Appendix A.

Table 3 presents the adjusted least-square mean estimates of the PCE scores (and SEs) and adjusted mean differences in the PCEs (and 95% CIs) according to rural–urban residence from separate analyses for the treated and untreated. The adjusted mean differences are beta coefficients from the linear regressions representing the mean differences in ratings or composite scores comparing rural and metro vs. large metro areas. Among those treated, rural PCa survivors (compared to large metro) PCa survivors on average recorded a 3.6-point higher score (95% CI: 0.5 to 6.7, *p* = 0.022) for getting needed care. However, among the untreated, rural PCa survivors reported significantly lower scores for obtaining needed care (ß = −7.0, 95% CI −12.7 to −1.2; *p* = 0.017) and a significantly lower rating for their health plan (ß = −7.9, 95% CI −13.1 to −2.6; *p* = 0.003) as compared to large metro residents. Metro PCa survivors recorded similar care experiences to large metro PCa survivors with no statistically significant differences in any of the PCEs in both the treated and untreated PCa survivors.

### 3.3. Startified Analyses by Treatment Status

Figure 2a,b present the unadjusted differences according to rural–urban status in receipt of definitive treatment and time to treatment, respectively, among those treated and stratified according to geographic regions. Rural–urban disparities in receipt of definitive treatment were only significant in the West region (*p* < 0.001) where a large proportion of rural PCa survivors (30.8%) did not receive definitive treatment compared to metro (15.3%) and large metro (17.4%) PCa survivors, respectively. No statistically significant differences were found in receipt of definitive treatment according to rural–urban residence in other regions (Figure 2a).

Figure 2b shows time to treatment according to rural–urban residence for different regions among the treated PCa survivors. Time to treatment differed significantly according to rural–urban residence in the South (*p* = 0.010) and West (*p* < 0.001) regions. In both regions, treated rural and metro PCa survivors were more likely to have received their treatment within one month than large metro (rural vs. metro vs. large metro: South region—46.5% vs. 38.2% vs. 32.3%; West region—34.7% vs. 38.9% vs. 29.2%) and less likely to have to wait at least 3 months before receiving their treatment (rural vs. metro vs. large metro: South region—14.7% vs. 12.6% vs. 23.3%; West region—18.8% vs. 15.5% vs. 23.3%). No statistically significant differences in time to treatment according to rural–urban residence in the Northeast (*p* = 0.480) and Midwest (*p* = 0.345) regions were observed.

## 4. Discussion

In this study, we explored rural–urban disparities in PCEs among intermediate/high-risk PCa survivors who did and did not receive definitive treatment. Previous studies evaluating PCEs among cancer survivors included rural–urban residence as a covariate [33,34,35], while some others included both rural–urban residence and treatment status as covariates [36,37]. However, none of the studies explored interactions between rural–urban residence and treatment status and their association with PCEs. Since receipt of treatment is associated with both rural–urban residence [7,8,9] and PCEs [37], treatment status may potentially moderate the associations between rural–urban residence and PCEs. Therefore, we also separately analyzed treated and untreated PCa survivors and found significant differences in rural–urban disparities in PCEs according to treatment status. While rural–urban residence was not significantly associated with PCEs when analyzing treated and untreated PCa survivors together, we observed significant associations of rural–urban residence with PCEs when we separately analyzed treated and untreated PCa survivors. However, these associations went in opposite directions in treated and untreated PCa survivors. Among untreated PCa survivors, rural (compared to large metro) PCa survivors reported significantly lower scores for obtaining needed care and a lower rating for their health plan. On the other hand, among treated PCa survivors, rural (compared to large metro) PCa survivors on average recorded a higher score for obtaining needed care. The opposing direction of associations in these two strata highlighted the importance of conducting a subgroup analysis to uncover any important heterogeneous effects among cancer survivors.

In the combined analyses of treated and untreated PCa survivors, lower average scores and ratings for all PCE measures according to untreated compared to treated PCa survivors suggested an overall poor care quality among the untreated. Although undergoing definitive treatment carries a risk of adverse long-term effects such as urinary incontinence, urinary obstruction, and hindered bowel and sexual functions that deteriorate the health-related quality of life [24], for PCa survivors with intermediate/high risk, the benefits of definitive treatment may outweigh the potential harm. Receipt of definitive treatment has been shown to significantly improve disease-specific and overall survival among PCa survivors at intermediate/high risk [38,39]. Moreover, the 2017 American Urological Association (AUA)/American Society for Radiation Oncology (ASTRO)/Society of Urologic Oncology (SUO) guidelines recommend definitive treatment for intermediate/high-risk PCa [6]. Doctor recommendations play a crucial role in PCa survivors deciding to obtain definitive treatment [40]. Poor patient–provider communication in such situations may negatively influence a patient’s decision to undergo definitive treatment. Our finding of lower average scores for doctor communication and a lower average specialist rating among untreated PCa survivors compared to treated PCa survivors may suggest that there might have been non-alignment of treatment recommendations with patient expectations regarding treatment outcomes.

Among treated PCa survivors, a comparatively higher proportion of rural (vs. large metro) PCa survivors received definitive treatment within one month of diagnosis. Needed care should be available promptly to achieve the best health outcomes [41]. The perception of definitive treatment as being “needed care” and accessing it in a timely manner may have resulted in treated rural PCa survivors recording higher scores for obtaining needed care as compared to treated large metro PCa survivors and engendering rural–urban disparities where urban PCa survivors were doing worse than rural PCa survivors.

On the other hand, we also observed that compared to large metro PCa survivors, a significantly higher proportion of rural PCa survivors were untreated, which was consistent with previous studies [42,43]. The untreated rural PCa survivors on average reported significantly lower scores for obtaining needed care and a lower rating for health plans as compared to their large metro counterparts. Although the actual reasons behind these disparities could not be directly inferred from this study, untreated PCa survivors may be managed through active surveillance, which involves actively monitoring the tumor and determining the future course of treatment as required [44]. Active surveillance generally involves long-term follow-up through biopsy, imaging, and PSA testing [45], which would require frequent visits to healthcare facilities; this may be more challenging for rural cancer survivors who may require long-distance travel and may delay or forego needed care to avoid traveling long distances [46]. Access to care for rural patients could be improved by implementing federal-level policies such as strengthening the network of critical access hospitals, improving rural healthcare infrastructure, and expanding the rural healthcare workforce. Moreover, incentivizing collaborations between urban centers and rural providers could help bridge gaps in providing specialty care to disadvantaged populations. Ensuring adherence to quality care delivery practices across all healthcare facilities may help distribute some of the patient influx from the large metros to the surrounding metros, thus potentially decreasing treatment wait times.

The bipolar nature of healthcare access among the rural PCa survivors, for which treated survivors reported better access to care compared to treated large metro survivors while untreated rural survivors fared worse in accessing care than untreated large metro survivors, suggested that possible variations exist among rural communities. To further explore this difference, we stratified rural–urban areas according to geographic regions and evaluated regional differences in receiving and time to receiving definitive treatment. We found significant rural–urban differences in time to receiving definitive treatment in the West and South regions but not in the Midwest and Northeast regions, which suggested that rural communities within different geographic areas may have differential access to timely PCa treatment. While this was consistent with previous research that also pointed out that rural–urban comparisons may reveal inconsistent patterns of disparity across the different geographical regions [47,48], larger studies are needed to confirm this finding and identify the rural communities that experience the greatest number of barriers to accessing care.

### Limitations

Several study limitations are worth noting when interpreting the study findings. The study sample consisted of PCa survivors who were Medicare enrollees and who completed a CAHPS survey after diagnosis (or treatment if they received definitive treatment), and the estimates were not weighted. Therefore, the study results may not be generalized to all Medicare enrollees. The PCEs reflected on the care received within the last 6 months of completing the survey. Although we used the first survey filled at least 6 months after diagnosis, >60% of the sample filled the first survey at least 2 years after diagnosis. Thus, the care experiences captured may or may not specifically pertain to the definitive treatment phase of care. We used information from the first course of treatment reported in SEER, which may under-report radiation treatment [49]. Further, regarding radiation, we did not have information on whether the provider recommended the treatment or why the patient refused radiation. The geographic variations in rural–urban disparities in receipt of definitive treatment and time to receiving definitive treatment were limited by sample size, especially in the Northeast region. Larger sample sizes are needed to confirm the findings.

## 5. Conclusions

To conclude, this study examined the rural–urban differences in PCEs of PCa cancer survivors with intermediate or high risk for disease progression. Compared to PCa survivors in large metro areas, we found rural PCa survivors were less likely to receive definitive treatment, but if they received it, they were more likely to have received it earlier. Moreover, we found bipolar associations of rural–urban residence with PCEs according to treatment status; treated rural PCa survivors reported better access to care, while untreated rural PCa survivors reported poorer access and experiences compared to their large metro counterparts. We also found some indication that rural–urban disparity in access to PCa treatment may differ by geographic region. These findings highlighted the complexity of the rural–urban disparities in PCEs and the importance of conducting a subgroup analysis to uncover any important heterogeneous effects among cancer survivors. Future larger studies are needed to confirm our findings and identify communities that experience the greatest number of barriers in accessing PCa care.

## Figures and Tables

**Figure 1 cancers-15-01939-f001:**
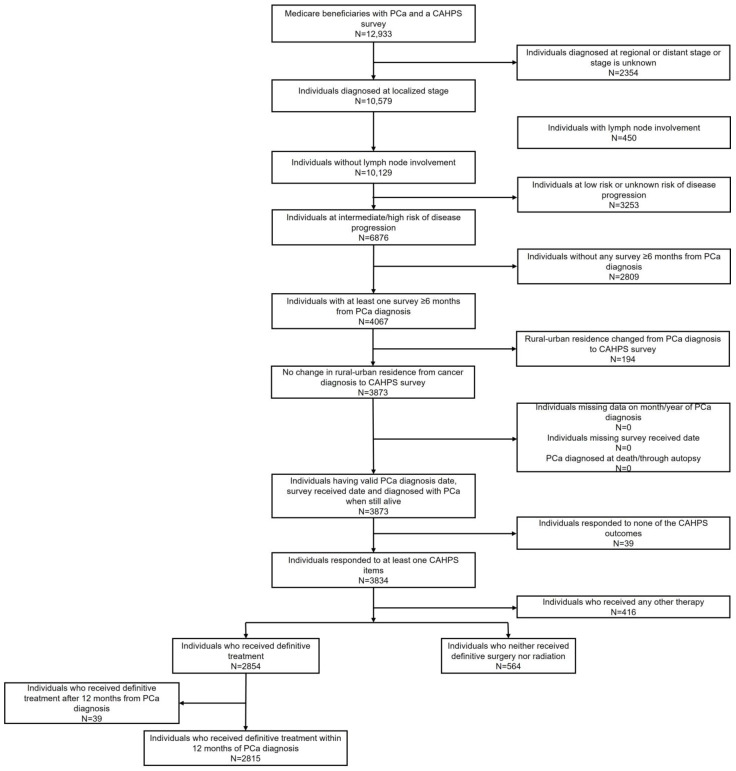
Sample selection flow chart. PCa—prostate cancer; CAHPS—Consumer Assessment of Healthcare Providers and Systems.

**Figure 2 cancers-15-01939-f002:**
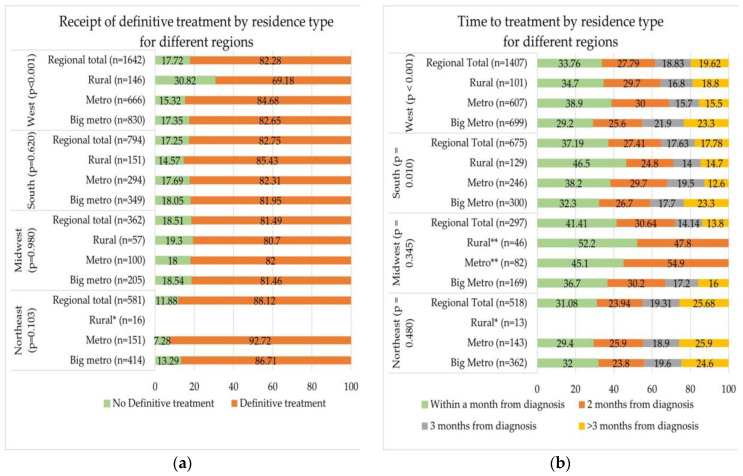
Disparities in (**a**) receipt of definitive treatment according to residence among different regions and (**b**) time to treatment disparities according to residence among different regions. * Cell sizes for each category <11 were suppressed per the Center for Medicare and Medicaid Services (CMS) Cell Size Suppression Policy to protect the confidentiality of respondents. ** Orange color represents the combined data for categories ‘2 months from diagnosis’, ‘3 months from diagnosis’, and ‘>3 months from diagnosis’ because each individual category had sample size < 11. This is done to protect the privacy of respondents per the CMS Cell Size Suppression Policy.

**Table 1 cancers-15-01939-t001:** Demographic and clinical characteristics of prostate cancer survivors according to residence type.

Variable	Large Metro (*n* = 1798)	Metro (*n* = 1211)	Rural (*n* = 370)	*p*-Value ^a^
**Age when responding to survey**	74.4 ± 6.7	74.6 ± 6.3	74.0 ± 6.1	0.371
**Time between prostate cancer diagnosis and CAHPS survey**				0.905
Less than 2 years	657 (36.5)	462 (38.2)	133 (36.0)	
2–5 years	825 (45.9)	542 (44.8)	172 (46.5)	
>5 years	316 (17.6)	207 (17.1)	65 (17.6)	
**Health plan type at the time of CAHPS survey**				<0.001
Medicare Advantage	830 (46.2)	557 (46)	229 (61.9)	
Fee-for-service	968 (53.8)	654 (54)	141 (38.1)	
**Race**				<0.001
Non-Hispanic White	1224 (68.1)	870 (71.8)	>290 (>78.4)	
Non-Hispanic Black	272 (15.1)	126 (10.4)	38 (10.3)	
Hispanic	171 (9.5)	75 (6.2)	15 (4.1)	
Non-Hispanic Asian	88 (4.9)	87 (7.2)	<11 (<3.0)	
Other	43 (2.4)	53 (4.4)	16 (4.3)	
**Education level**				<0.001
Some college or above	1069 (59.5)	687 (56.7)	170 (46.0)	
High School or less	637 (35.4)	455 (37.6)	184 (49.7)	
Missing	92 (5.1)	69 (5.7)	16 (4.3)	
**Proxy answered questions for the respondent**				0.586
No	1402 (78.0)	915 (75.6)	284 (76.8)	
Yes	160 (8.9)	114 (9.4)	32 (8.7)	
Missing	236 (13.1)	182 (15)	54 (14.6)	
**Low-income subsidy**				0.084
No	1566 (87.1)	1087 (89.8)	325 (87.8)	
Yes	232 (12.9)	124 (10.2)	45 (12.2)	
**Dual eligibility for Medicare and Medicaid**				0.007
No	1590 (88.4)	1106 (91.3)	329 (88.9)	
Yes	195 (10.9)	>94 (>7.8)	>30 (>8.1)	
Missing	13 (0.7)	<11 (<0.9)	<11 (<3.0)	
**Marital status**				0.271
Not Married	377 (21.0)	220 (18.2)	72 (19.5)	
Married	1184 (65.9)	841 (69.5)	255 (68.9)	
Missing	237 (13.2)	150 (12.4)	43 (11.6)	
**Geographic region when answering the CAHPS survey**				<0.001
Northeast	414 (23.0)	151 (12.5)	16 (4.3)	
Midwest	205 (11.4)	100 (8.3)	57 (15.4)	
South	349 (19.4)	294 (24.3)	151 (40.8)	
West	830 (46.2)	666 (55.0)	146 (39.5)	
**Census tract poverty indicator**				<0.001
0–<5% poverty	585 (32.5)	276 (22.8)	28 (7.6)	
5% to <10% poverty	517 (28.8)	328 (27.1)	69 (18.7)	
10% to <20% poverty	421 (23.4)	386 (31.9)	160 (43.2)	
20% to 100% poverty	>264 (>14.7)	>210 (>17.3)	>102 (>27.6)	
Missing	<11 (<0.6)	<11 (<0.9)	<11 (<3.0)	
**Smoking status**				0.074
Non-Smoker	1532 (85.2)	1029 (85.0)	303 (81.9)	
Smoker	164 (9.1)	122 (10.1)	51 (13.8)	
Missing	102 (5.7)	60 (5.0)	16 (4.3)	
**Survey year**				0.018
2008	65 (3.6)	35 (2.9)	12 (3.2)	
2009	136 (7.6)	95 (7.8)	32 (8.7)	
2010	193 (10.7)	124 (10.2)	25 (6.8)	
2011	218 (12.1)	159 (13.1)	54 (14.6)	
2012	274 (15.2)	238 (19.7)	73 (19.7)	
2013	316 (17.6)	219 (18.1)	60 (16.2)	
2014	309 (17.2)	196 (16.2)	65 (17.6)	
2015	287 (16.0)	145 (12.0)	49 (13.2)	
**Comorbidity count**				0.816
0	778 (43.3)	545 (45.0)	166 (44.9)	
1	629 (35.0)	414 (34.2)	116 (31.4)	
2	300 (16.7)	193 (15.9)	68 (18.4)	
3 or 4	91 (5.1)	59 (4.9)	20 (5.4)	
**Tumor grade**				0.020 *
Well differentiated	<11 (<0.6)	<11 (<0.9)	<11 (<3.0)	
Moderately differentiated	387 (21.5)	300 (24.8)	76 (20.5)	
Poorly differentiated	>1338 (>74.4)	>868 (>71.7)	>255 (>68.9)	
Undifferentiated	<11 (<0.6)	<11 (<0.9)	<11 (<3.0)	
Unknown	51 (2.8)	21 (1.7)	17 (4.6)	
**Time between prostate cancer diagnosis and receiving definitive treatment**				<0.001
Within a month from diagnosis	469 (26.1)	394 (32.5)	122 (33)	
2 months from diagnosis	387 (21.5)	294 (24.3)	77 (20.8)	
3 months from diagnosis	297 (16.5)	174 (14.4)	41 (11.1)	
>3 months from diagnosis	345 (19.2)	166 (13.7)	49 (13.2)	
Never	300 (16.7)	183 (15.1)	81 (21.9)	
**Receipt of radiation as a part of initial treatment**				0.342 *
No	>858 (>47.7)	>603 (>49.8)	>187 (>50.5)	
Yes	929 (51.7)	597 (49.3)	172 (46.5)	
Missing	<11 (<0.6)	<11 (<0.9)	<11 (<3.0)	
**Receipt of definitive surgery as a part of initial treatment**				0.091 *
No	1218 (67.7)	>763 (>63.0)	249 (67.3)	
Yes	580 (32.3)	437 (36.1)	121 (32.7)	
Missing	0 (0.0)	<11 (<0.9)	0 (0.0)	
**Risk of disease progression**				0.538
Intermediate	1333 (74.1)	894 (73.8)	264 (71.4)	
High	465 (25.9)	317 (26.2)	106 (28.7)	
**Number of prior cancers other than prostate cancer**				0.113
0	1611 (89.6)	1101 (90.9)	339 (91.6)	
1	171 (9.5)	>83 (>6.9)	>20 (>5.4)	
2	16 (0.9)	16 (1.3)	<11 (<3.0)	
3	0 (0)	<11 (<0.9)	0 (0)	
**General health status**				0.516
Missing	51 (2.8)	42 (3.5)	13 (3.5)	
Excellent	153 (8.5)	94 (7.8)	26 (7.0)	
Very good	520 (28.9)	351 (29.0)	100 (27.0)	
Good	679 (37.8)	463 (38.2)	131 (35.4)	
Fair	336 (18.7)	216 (17.8)	80 (21.6)	
Poor	59 (3.3)	45 (3.7)	20 (5.4)	
**Mental health status**				0.061
Missing	55 (3.1)	38 (3.1)	10 (2.7)	
Excellent	610 (33.9)	381 (31.5)	97 (26.2)	
Very good	563 (31.3)	400 (33.0)	143 (38.7)	
Good	409 (22.8)	299 (24.7)	83 (22.4)	
Fair	135 (7.5)	83 (6.9)	29 (7.8)	
Poor	26 (1.5)	10 (0.8)	8 (2.2)	

Column percentages may not sum to 100.0% due to rounding. ^a^ ANOVA was used for the continuous variable (age when responding to survey); chi-square or Fisher’s exact test was used as appropriate for categorical variables. * Cells with sizes < 11 and one other cell for the same variable were suppressed per the Center for Medicare and Medicaid Services (CMS) Cell Size Suppression Policy to protect the confidentiality of respondents.

**Table 2 cancers-15-01939-t002:** Least-square mean estimates and adjusted differences in patient experience measures according to treatment status.

Variable	Treated LSM ± SE	Not Treated LSM ± SE	Adjusted Difference (95% CI)
**Composite measures**			
Obtaining needed care (*n* = 2460) Obtaining care quickly (*n* = 2864) Doctor communication (*n* = 2504)	85.3 ± 6.2	85.2 ± 6.2	−0.1 (−2.4 to 2.2)
	61.1 ± 6.0	59.6 ± 6.0	−1.4 (−4.1 to 1.2)
	86.9 ± 4.9	84.9 ± 4.9	−2.0 (−3.8 to −0.2) *^,+^
Obtaining needed prescription drugs (*n* = 1811)	93.4 ± 6.2	92.3 ± 6.3	−1.1 (−3.7 to 1.6)
Customer service (*n* = 878)	68.5 ± 8.7	65.2 ± 8.8	−3.3 (−8.2 to 1.6)
**Ratings**			
Primary care provider rating (*n* = 2494)	88.2 ± 4.4	86.9 ± 4.4	−1.4 (−2.9 to 0.2)
Specialist rating (*n* = 2089)	87.4 ± 5.1	84.9 ± 5.1	−2.5 (−4.4 to −0.7) *^,+^
Health plan rating (*n* = 2956)	85.2 ± 4.5	83.6 ± 4.5	−1.6 (−3.5 to 0.3)
Overall care rating (*n* = 2913)	89.0 ± 4.1	86.6 ± 4.1	−2.4 (−4.1 to −0.7) *^,+^

LSM—least-square mean; SE—standard error; CI—confidence interval. * Significant at *p* < 0.05. ^+^ Small difference; ^++^ medium difference; ^+++^ large difference [26]. Only statistically significant differences were categorized as small, medium, and large differences.

**Table 3 cancers-15-01939-t003:** Least-square mean estimates and adjusted differences in patient experience measures according to residence types and stratified by treatment status.

Variable	Large Metro LSM ± SE	Metro LSM ± SE	Metro vs. Large Metro Adjusted Difference (95% CI)	Rural LSM ± SE	Rural vs. Large Metro Adjusted Difference (95% CI)
	**Treated**				
**Composite measures**					
Obtaining needed care (*n* = 2460)	82.8 ± 6.9	83.8 ± 6.9	1 (−0.8 to 2.8)	86.4 ± 7.0	3.6 (0.5 to 6.7) *^,++^
Obtaining care quickly (*n* = 2864)	58.6 ± 6.3	59.1 ± 6.3	0.5 (−1.6 to 2.6)	60.7 ± 6.5	2.1 (−1.5 to 5.7)
Doctor communication (*n* = 2504)	86.6 ± 5.3	86.9 ± 5.3	0.3 (−1.1 to 1.7)	88.7 ± 5.4	2.1 (−0.3 to 4.4)
Obtaining needed prescription drugs (*n* = 1811)	90.7 ± 7.1	92.7 ± 7.0	2 (−0.1 to 4.1)	92.7 ± 7.2	2 (−1.7 to 5.7)
Customer service (*n* = 716)	64.5 ± 10.1	61.7 ± 10.2	−2.8 (−6.8 to 1.3)	63.7 ± 10.8	−0.8 (−8.4 to 6.8)
**Ratings**					
Primary care provider rating (*n* = 2494)	87.0 ± 4.8	87.5 ± 4.8	0.4 (−0.8 to 1.7)	87.9 ± 4.9	0.9 (−1.2 to 3)
Specialist rating (*n* = 2089)	84.0 ± 4.2	84.3 ± 4.2	0.3 (−1 to 1.7)	84.5 ± 4.3	0.6 (−1.8 to 2.9)
Health plan rating (*n* = 2956)	83.0 ± 4.7	82.3 ± 4.7	−0.7 (−2.2 to 0.8)	83.1 ± 4.8	0.1 (−2.5 to 2.7)
Overall care rating (*n* = 2913)	88.6 ± 4.3	88.6 ± 4.2	0 (−1.3 to 1.3)	89.4 ± 4.4	0.8 (−1.4 to 3.1)
	**Not Treated**				
**Composite measures**					
Obtaining needed care (*n* = 391)	86.1 ± 8.0	82.2 ± 8.0	−3.9 (−8.2 to 0.4)	79.1 ± 8.4	−7.0 (−12.7 to −1.2) *^,+++^
Obtaining care quickly (*n* = 474)	62.0 ± 10.4	64.5 ± 10.4	2.5 (−2.7 to 7.7)	58.2 ± 10.9	−3.9 (−10.9 to 3.2)
Doctor communication (*n* = 411)	78.0 ± 7.2	76.9 ± 7.3	−1.1 (−4.7 to 2.5)	78.2 ± 7.6	0.2 (−4.7 to 5.2)
Obtaining needed prescription drugs (*n* = 323)	78.2 ± 8.9	76.3 ± 9.0	−1.9 (−7.2 to 3.4)	80.1 ± 9.2	1.9 (−5.3 to 9.1)
Customer service (*n* = 162)	59.0 ± 12.0	60.9 ± 11.0	1.9 (−7.9 to 11.6)	53.6 ± 13	−5.4 (−19.5 to 8.6)
**Ratings**					
Primary care provider rating (*n* = 404)	81.5 ± 5.9	80.2 ± 5.9	−1.3 (−4.3 to 1.6)	79.7 ± 6.2	−1.8 (−5.8 to 2.3)
Specialist rating (*n* = 333)	84.1 ± 8.0	83.2 ± 7.9	−0.9 (−5.2 to 3.5)	80.5 ± 8.4	−3.7 (−9.6 to 2.3)
Health plan rating (*n* = 498)	90.4 ± 8.0	87.5 ± 8.0	−3.0 (−6.8 to 0.9)	82.6 ± 8.3	−7.9 (−13.1 to −2.6) *^,+++^
Overall care rating (*n* = 476)	83.0 ± 7.4	82.7 ± 7.4	−0.3 (−4.1 to 3.5)	78.2 ± 7.7	−4.8 (−10 to 0.4)

LSM—least-square mean; SE—standard error; CI: confidence interval. * Significant at *p* < 0.05. ^+^ Small difference; ^++^ medium difference; ^+++^ large difference [26]. Only statistically significant differences were categorized as small, medium, and large differences.

## Data Availability

The SEER-CAHPS data used to conduct this study can be acquired from the National Cancer Institute upon request.

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
