# Peer review of "Rural–Urban Disparities in Patient Care Experiences among Prostate Cancer Survivors: A SEER-CAHPS Study"

_cancers, 2023, doi:10.3390/cancers15071939_

Round 1

Reviewer 1 Report

Unfortunately, the place of residence still determines the availability of medical services. The presented analysis is very important. It gives evidence of the inequality of patient rights. However, the summary lacks recommendations for the past. The result of the study should bring more than just knowledge. Decision makers should know how to change this. Authors should describe which variables are modifiable.

Author Response

General response

All the references provided for line numbers, tables and figures are from the ‘All markup’ version of the revised document.

Reviewer 1 comments

Unfortunately, the place of residence still determines the availability of medical services. The presented analysis is very important. It gives evidence of the inequality of patient rights. However, the summary lacks recommendations for the past. The result of the study should bring more than just knowledge. Decision makers should know how to change this. Authors should describe which variables are modifiable.

Response: Thank you for the comment. We have revised the manuscript on lines 331-338 to read "Access to care for rural patients could be improved by implementing federal level policies such as strengthening the network of critical access hospitals, improving rural healthcare infrastructure, and expanding the rural healthcare work force. Moreover, incentivizing collaborations between urban centers and rural providers could help bridge gaps in providing specialty care to disadvantaged populations. Ensuring adherence to quality care delivery practices across all healthcare facilities may help distribute some of the patient influx from the big metros to the surrounding metros, thus potentially decreasing treatment wait times.”

Reviewer 2 Report

The Interesting topic discussed is threatened by some imperfections such as the lack of information about PCa epidemiology and risk factors: these papers (https://doi.org/10.3390%2Fijerph18168500https://doi.org/10.3390%2Fdiagnostics11050908  offer the most up-to-date details about them and would guarantee a higher scientific impact to the work. 

Methods and methodology are robust.

The “Figure 1” definition could be implemented.

I believe that the study has sufficient merit to be considered for publication, although major revisions are required. 

Author Response

General response

All the references provided for line numbers, tables and figures are from the ‘All markup’ version of the revised document.

Reviewer 2 comments

  1. The Interesting topic discussed is threatened by some imperfections such as the lack of information about PCa epidemiology and risk factors: these papers (https://doi.org/10.3390%2Fijerph18168500, https://doi.org/10.3390%2Fdiagnostics11050908) offer the most up-to-date details about them and would guarantee a higher scientific impact to the work.

Response: We thank the reviewer for providing useful references to cite in our manuscript. We have revised the manuscript on lines 53-55 to read “Globally, 1,276,000 new PCa cases were diagnosed, and 359,000 men died from PCa in 2018 [2]. A positive family history of PCa, besides age and African American ancestry, is the most significant risk factor for developing PCa [3].”

Reference [2] from the manuscript: Crocetto F, Arcaniolo D, Napolitano L, Barone B, La Rocca R, Capece M, Caputo VF, Imbimbo C, De Sio M, and Calace FP. Impact of Sexual Activity on the Risk of Male Genital Tumors: A Systematic Review of the Lit-erature. International Journal of Environmental Research and Public Health. 2021;18(16):8500-8515. 10.3390/IJERPH18168500.

Reference [3] from the manuscript: Crocetto F, Barone B, Caputo VF, Fontana M, de Cobelli O, and Ferro M. BRCA Germline Mutations in Prostate Cancer: The Future Is Tailored. Diagnostics. 2021;11(5):908-911. 10.3390/DIAGNOSTICS11050908.

  1. The “Figure 1” definition could be implemented.

Response: We have provided figure 1 with a better resolution in the revised manuscript and also provided a JPEG copy as an attachment.

Reviewer 3 Report

In this article, Pandit and colleagues compare the experience of prostate cancer diagnosis between men living in large urban areas and men living in rural areas.

Rural residency is known to be associated with poor access to quality healthcare services due to sub trained urologists, hard access to advanced MRIs, etc…

In this study, the authors evaluated disparities between rural-urban men among patients with intermediate/high-risk prostate cancer in terms of receiving definitive treatment.

They observed that for untreated patients, rural survivors experienced significantly lower needs and quality-perceived care than patients in big-metro. This difference has not been accounted for in past works on health disparities.

According to other studies, the also observed that a significantly higher percentage of rural patients survivors were untreated.

A very interesting work that opens up important points of reflection to improve the quality of life of patients with prostate cancer.

At the end of the discussion, the authors hypothesize that there is a further difference in access to health services not only based on the urban-rural area, but also on the geographical location (north, south, east, west).

This aspect is particularly interesting.

Are there studies on other pathologies (for example) that could support this hypothesis?

Author Response

General response

All the references provided for line numbers, tables and figures are from the ‘All markup’ version of the revised document.

Reviewer 3 comments

In this article, Pandit and colleagues compare the experience of prostate cancer diagnosis between men living in large urban areas and men living in rural areas.

Rural residency is known to be associated with poor access to quality healthcare services due to sub trained urologists, hard access to advanced MRIs, etc…

In this study, the authors evaluated disparities between rural-urban men among patients with intermediate/high-risk prostate cancer in terms of receiving definitive treatment.

They observed that for untreated patients, rural survivors experienced significantly lower needs and quality-perceived care than patients in big-metro. This difference has not been accounted for in past works on health disparities.

According to other studies, the also observed that a significantly higher percentage of rural patients survivors were untreated.

Response: Thank you for your comments.

A very interesting work that opens up important points of reflection to improve the quality of life of patients with prostate cancer. At the end of the discussion, the authors hypothesize that there is a further difference in access to health services not only based on the urban-rural area, but also on the geographical location (north, south, east, west). This aspect is particularly interesting. Are there studies on other pathologies (for example) that could support this hypothesis?

Response: Thank you for the comment. In the chapter 5 of the book titled ‘Guidance for the National Healthcare Disparities Report’, Rickets TC III has provided examples of how geographic location, in addition to rural-urban area, may be associated disparities in access to health services.

https://www.ncbi.nlm.nih.gov/books/NBK221045/

Similarly, a systematic literature review by Cyr et al., also provides several examples where rural-urban disparities in healthcare access vary by the state, which serves as a smaller component of the geographic locations categorized as Northeast, Midwest, South, and West.

https://bmchealthservres.biomedcentral.com/articles/10.1186/s12913-019-4815-5

We have revised our manuscript on line 348-351 to read “While this is consistent with previous research that also points out that rural-urban comparisons may reveal inconsistent patterns of disparity across the different geographical regions [47, 48], larger studies are needed to confirm this finding and identify the rural communities that experience the most barriers in accessing care.”

Reference [47] from the manuscript: Thomas C. Ricketts III. (2002). Geography and disparities in health care. In Elaine K. Swift (Ed.), Guidance for the National Healthcare Disparities Report (1st ed., p. 216). National Academies Press (US). Retrieved from https://www.ncbi.nlm.nih.gov/books/NBK221045/

Reference [48] in the manuscript: Cyr, M. E., Etchin, A. G., Guthrie, B. J., & Benneyan, J. C. (2019). Access to specialty healthcare in urban versus rural US populations: A systematic literature review. BMC Health Services Research, 19(1), 1–17. https://doi.org/10.1186/S12913-019-4815-5/FIGURES/4

Round 2

Reviewer 2 Report

Authors answered all comments and suggestions.